# Disclosure of conventional and complementary medicine use to medical doctors and complementary medicine practitioners: A survey of rates and reasons amongst those with chronic conditions

**Hope Foley**[1]*, **Amie Steel**[1], **Erica McIntyre**[1], **Joanna Harnett**[1,2], **David Sibbritt**[1], **Jon Adams**[1]

**1** Australian Research Centre in Complementary and Integrative Medicine, Faculty of Health, University of Technology Sydney, Ultimo, New South Wales, Australia, **2** School of Pharmacy, Faculty of Medicine and Health, The University of Sydney, Sydney, New South Wales, Australia

* hope.m.foley@student.uts.edu.au

**Data Availability Statement:** Our data can be made available upon request but cannot be lodged

## Abstract

Chronic conditions are prolonged and complex, leading patients to seek multiple forms of care alongside conventional treatment, including complementary medicine (CM). These multiple forms of care are often used concomitantly, requiring patient-provider communication about treatments used in order to manage potential risks. In response, this study describes rates and reasons for disclosure/non-disclosure of conventional medicine use to CM practitioners, and CM use to medical doctors, by individuals with chronic conditions. A survey was conducted online in July and August 2017 amongst the Australian adult population. Participants with chronic conditions were asked about their disclosure-related communication with CM practitioners (massage therapist, chiropractor, acupuncturist, naturopath) and medical doctors. Patients consulting different professions reported varying disclosure rates and reasons. Full disclosure (disclosed ALL) to medical doctors was higher (62.7%-79.5%) than full disclosure to CM practitioners (41.2%-56.9%). The most strongly reported reason for disclosing to both MDs and CM practitioners was *I wanted them to fully understand my health status*, while for non-disclosure it was *They did not ask me about my CM/medicine use*. Reasons regarding concerns or expectations around the consultation or patient-provider relationship were also influential. The findings suggest that patient disclosure of treatment use in clinical consultation for chronic conditions may be improved through patient education about its importance, direct provider inquiry, and supportive patient-provider partnerships. Provision of optimal patient care for those with chronic conditions requires greater attention to patient-provider communication surrounding patients' wider care and treatment use.

into a public repository as the ethics approval received for the study does not allow for public sharing of this nature. Due to the survey containing health-related data, the Human Research Ethics Committee (HREC) at University of Technology Sydney (UTS) did not include public sharing of the dataset within the approval specifications. The UTS HREC can be contacted at (+61) 2 9514 9772, or data access can also be requested via the research team (erica.mcintyre@uts.edu.au).

**Funding:** This study was supported by direct contributions from Endeavour College of Natural Health and the Australian Research Centre in Complementary and Integrative Medicine (ARCCIM), University of Technology Sydney. HF was supported by an Australian Government Research Training Program Scholarship while working on this manuscript. JA was supported by an Australian Research Council Professorial Future Fellowship while working on this manuscript (Grant FT140100195). JH's academic position was supported by a philanthropic donation from Blackmores Pty Ltd during the course of this study. Funding sources had no role in the study design, data collection or analysis, decision to publish, or preparation of the manuscript. No commercial interests were associated with the study or its outcomes.

**Competing interests:** The authors have declared that no competing interests exist.

## Introduction

The increasing prevalence of chronic conditions over recent decades is due to the culmination of many factors including advances in medical treatment of infectious diseases, an ageing population, and post-industrial changes to dietary and lifestyle habits less conducive to health maintenance [1, 2]. Health systems must adapt to address the substantial medical and economic burden of chronic conditions, and to meet the different needs associated with chronic conditions for affected patients [3]. Chronic conditions affect the functional capacity of individuals over a protracted course of time and often involve multiple predisposing, precipitating and perpetuating factors [4]. Such complexity often leads to reduced quality of life, social and socioeconomic impacts on individuals, families and communities, and a need for continuous, ongoing provision of medical care accounting for both direct and indirect outcomes of chronic conditions [4].

Those living with chronic conditions often seek a multi-focused approach to treatment management including use of both conventional/pharmaceutical medicine and complementary medicine (CM) [5]. CM is a field encompassing those health and medical practices and products that are separate from mainstream medical systems, practice and education [6]. CM may include self-prescribed products and practices, or care provided by practitioners of CM professions [6], and individuals with chronic conditions use CM at higher rates than the general population [7]. While concomitant CM and conventional medicine use may be customised to help address the broad and diverse needs of those living with chronic a condition(s) [8], there are also potential risks involved, such as interactions between different medicines/treatments, or use of medicines/treatments that may be contraindicated or unnecessary in the presence of certain chronic conditions [9]. In order to ensure potential risks are avoided or appropriately managed, it is important for patients and care providers to communicate about the treatments being used [10].

Previous literature has examined patient disclosure of CM use to conventional medical providers (e.g. medical doctors, pharmacists, nurses) within the general population and while findings vary across studies, rates of disclosure are on average 33% [11]. The reasons patients report for non-disclosure often relate to a lack of inquiry from care providers, fear of disapproval from the provider, and a lack of understanding of the importance of disclosing CM use [11]. Conversely, patients who disclose their CM use to conventional medical providers often give their reasons for disclosing as being related to provider inquiry, belief they will be supported by their provider, and/or an understanding of the importance of disclosing [11]. Disclosure of conventional medicine use to CM practitioners has not yet been explored beyond a few preliminary studies which briefly report on rates of disclosure of conventional prescription medications to naturopaths [12] or CM practitioners more broadly [13]. These studies have yielded mixed results, suggesting patient disclosure behaviors to CM practitioners may vary across different settings, populations or demographic groups.

Despite this early work, the topic of medicine disclosure to care providers has not been subject to rigorous investigation within the clinical population of those with chronic conditions. Additionally, no validated instrument has been consistently implemented to examine disclosure rates or reasons in either complementary or conventional medicine settings to date. This study aimed to describe the rates of and reasons for disclosure and non-disclosure of conventional medicine use to CM practitioners, and of CM use to medical doctors (MDs), amongst individuals with chronic conditions, using novel, validated measures.

## Materials and methods

### Study design and setting

This paper reports on data collected via the Complementary and Alternative Medicine Use, Health Literacy and Disclosure (CAMUHLD) cross-sectional survey conducted online between 26 July and 28 August 2017. The survey was administered nationally across Australia [14]. Analyses presented here utilise data from a sub-set of the CAMUHLD sample.

### Participants and recruitment

Survey participants were adult members (aged 18 and over) of the Qualtrics research recruitment company's database, via which they were invited to participate. A sample broadly representative of the Australian population (regarding gender, age and state of residence) was achieved through employment of purposive convenience sampling. In line with Qualtrics operations, participants received a small financial recompense for their time as database members upon completion of the survey. Consent was provided by participants after reading an information sheet and the survey was approximately 15 minutes in length.

An initial sample of 2,025 participants was achieved in the CAMUHLD project. Six cases were removed due to discrepancies in responses that deemed the data unreliable, resulting in a project sample of 2,019. Analyses presented here utilise data regarding disclosure behaviors provided by respondents who: a) indicated having a chronic condition, and b) had consulted with CM practitioners from one of the professions most commonly accessed by respondents with chronic conditions (massage therapy, chiropractic, acupuncture and naturopathy). The final sample for the current analyses represents 302 participants.

### Instrument

The fifty-item CAMUHLD survey, administered online, included domains of socio-demographics, health status, health service utilisation, and health communication. Items utilised from socio-demographics covered gender, age, state of residence, financial manageability, level of education, employment status, relationship status, private health insurance (PHI) coverage, and possession of a Health Care Card (provided to low-income earners and welfare recipients in Australia for financial concessions on health care and medicines). Health status items covered diagnosis of or treatment for a chronic health condition within the preceding three years (participants were presented with a list of conditions as well as an open-text option). The health service utilisation items used included consultation within the preceding twelve-months with an acupuncturist, chiropractor, massage therapist, naturopath (CM practitioners), GP or specialist doctor (MDs).

Health communication items included initial questions that asked about rates of disclosure to each type of health professional consulted (*Disclosed ALL*, *Disclosed SOME*, *Did NOT disclose*). Participants were then presented with two novel measures, which were subsequently subjected to validation analyses after data collection: the Complementary Medicine Disclosure Index (CMDI; disclosure/non-disclosure of CM to conventional medical providers) [15], and the Conventional Medicine Disclosure Index (CONMED-DI; disclosure/non-disclosure of conventional medicine to CM practitioners) [16]. These indices each consisted of two lists of items measuring the reasons for: a) disclosure; and b) non-disclosure of the relevant medicine type, and were assessed with a five-point Likert scale ranging from *Strongly disagree* (1) to *Strongly agree* (5). Participants were directed to the CMDI (for consultations with MDs) or CONMED-DI (for consultations with CM practitioners) in accordance with the type of health professional they reported disclosing/not disclosing to. Those who indicated they had

*Disclosed ALL* were directed to the CMDI/CONMED-DI items for disclosure, participants who indicated they *Did NOT disclose* were directed to the CMDI/CONMED-DI items for non-disclosure, while participants who indicated they had *Disclosed SOME* were directed to both lists of items for the relevant index.

## Data analysis

Data analysis was undertaken using Stata-14 (StataCorp LC 2015) software. Categorical variables were recoded to produce binaries as necessary for analyses including health status (presence of chronic condition: yes/no) and health service utilisation (profession consulted: yes/no). Categorical variables outlining disclosure behaviors were also recoded to a binary for backward stepwise logistic regression analyses of potential predictors for full disclosure (disclosed all/did not disclose all). Independent variables were selected for inclusion in regression analysis through Pearson chi-square tests of association with disclosure; variables with a statistical significance of $p < 0.25$ were retained for the analysis. In order to preserve data integrity, responses to disclosure questions were only included in analysis if the respondent had indicated that they had consulted with a practitioner of the health profession being disclosed to within the previous twelve months. Complete responses were encouraged by the online survey layout, minimising missing data.

Frequencies and percentages were calculated for socio-demographic items and disclosure rates (disclosed ALL, disclosed SOME, or did NOT disclose), presented as sub-groups delineated by the health profession consulted. Chi-square analyses were used to test associations between respondents who did and did not consult with each of the four CM professions across socio-demographics and disclosure rates to MDs, with effect size determined by Cramer's V. Statistical significance was set at $p < 0.05$ and the effect size of associations was classified as negligible (under 0.10), weak (0.10 to under 0.20), moderate (0.20 to under 0.40), relatively strong (0.40 to under 0.60), strong (0.60 to under 0.80) or very strong (0.80 to 1.00). Potential socio-demographic predictors for having fully disclosed were explored through reverse stepwise logistic regression.

Reasons for disclosure and non-disclosure were calculated as means with standard deviation to estimate the relative importance of each reason, with higher means indicating stronger agreement with the item on average. Independent t-tests were used to assess differences in reasons between respondents who did and did not consult with each of the four CM professions. Levene's test was first applied to assess equality of variance. For variables which violated the assumption of equality of variance, Welch's t-test was employed.

## Ethics

The project conformed with the Declaration of Helsinki and received ethical approval from the Human Research Ethics Committee at Endeavour College of Natural Health (EC00358) (#20170242).

## Results

### Participant characteristics

Participants were most commonly female (n = 170, 56.3%), aged 18–29 years (n = 71, 32.5%), residing in the state of New South Wales (n = 88, 29.1%) and indicated that financial manageability was *difficult some of the time* (n = 114, 37.8%). Participants most commonly held trade/vocational (n = 105, 34.8%) or university (n = 104, 34.4%) qualifications, and were employed full-time (n = 101, 33.4%). Respondents were predominantly married (n = 142, 47.0%) and

held PHI cover (n = 186, 61.6%), with many having PHI for CM (n = 144, 47.7%). A majority of participants indicated possession of a HCC (n = 177, 58.6%).

Table 1 shows that massage therapists were consulted by 61.6% (n = 186) of respondents, chiropractors by 44.0% (n = 133), acupuncturists by 27.5% (n = 83) and naturopaths by 22.2% (n = 67). Compared to those consulting the other professions, having consulted a chiropractor was moderately associated with male gender (Cramer's V = 0.204, *p<0.001*), and having consulted a naturopath demonstrated a negligible association with full time employment (Cramer's V = 0.025, *p = 0.025*).

## Disclosure rates and their relation to CM profession consulted

Table 2 presents the rates of disclosure to CM practitioners, GPs and specialist doctors according to the CM profession consulted. Full disclosure of conventional medicine use (*Disclosed ALL*) to CM practitioners tended to be lower than rates of full disclosure of CM to MDs (GPs and specialist doctors). Overall, full disclosure rates were highest for disclosure of CM to specialist doctors. Accordingly, conventional medicine non-disclosure (*Did NOT disclose*) to CM practitioners tended to be higher than rates of CM non-disclosure to GPs and specialist doctors.

Respondents who had consulted a naturopath reported the highest rates of full disclosure of conventional medicines to CM practitioners (56.9%), followed closely by those who had consulted a chiropractor (56.8%). Respondents who had consulted a massage therapist had the highest rates of non-disclosure of conventional medicines to CM practitioners (35.6%).

The highest rates of full disclosure of CM to GPs were reported by respondents who had consulted a massage therapist (70.0%), while the highest rates of disclosure of CM to specialist doctors were reported by those who had consulted a chiropractor (79.5%). Respondents who had consulted an acupuncturist reported the highest rates of non-disclosure both to GPs (12.1%) and specialist doctors (12.0%). No statistically significant differences were seen in CM disclosure to MDs between respondents consulting with each of the four CM professions.

## Predictors of full disclosure

Backwards stepwise logistic regression models did not yield any predictive factors for full disclosure of conventional medicines to CM practitioners, or for full disclosure of CM to specialist doctors. However, full disclosure of CM to GPs was found to be predicted by age, financial manageability and number of chronic conditions. Respondents aged 50–59 years (AOR 3.51, *p = 0.004*, 95%CI 1.50, 8.20) and 60 and over (AOR 3.12, *p = 0.002*, 95%CI 1.52, 6.32) were found to have more than three times the odds of disclosing all CM use to their GPs. Respondents who indicated financial manageability as *difficult all of the time* had more than twice the odds of disclosing all CM to their GP (AOR 2.06, *p = 0.029*, 95%CI 1.08, 3.93). The odds of disclosing all CM to a GP increased with the number of chronic conditions, reaching statistical significance for those with four chronic conditions (AOR 2.63, *p = 0.021*, 95%CI 1.15, 5.99) and five or more chronic conditions (AOR 2.77, *p = 0.006*, 95%CI 1.35, 5.69).

## Reasons for disclosure and non-disclosure of CM use to MDs

Table 3 reports the reasons selected by participants who completed the CMDI for disclosure of CM use to MDs (n = 263), including results of independent t-tests exploring differences between those who did and did not consult with each type of CM professional. The the most agreement was indicated for the item *I wanted them to fully understand my health status* (mean = 4.44, SD = 0.73), followed by *I was concerned about drug interactions with the CM I was using* (mean = 4.20, SD = 0.89). The items that attracted the least agreement were *They*

**Table 1. Socio-demographic characteristics of participants and associations with complementary medicine professional consulted.**

| | Total sample n = 302 (100.0%) | Massage n = 186 (61.6%) | p-value[†] | Chiropractic n = 133 (44.0%) | p-value[†] | Acupuncture n = 83 (27.5%) | p-value[†] | Naturopathy n = 67 (22.2%) | p-value[†] |
|---|---|---|---|---|---|---|---|---|---|
| **Gender** | | | | | | | | | |
| Female | 170 (56.3%) | 114 (61.3%) | 0.310 | 59 (44.4%) | <0.001 (0.204) | 44 (53.0%) | 0.245 | 37 (55.2%) | 0.538 |
| Male | 132 (43.7%) | 72 (38.7%) | | 74 (55.6%) | | 39 (47.0%) | | 30 (44.8%) | |
| **Age** | | | | | | | | | |
| 18–29 | 71 (32.5%) | 45 (24.2%) | 0.113 | 38 (28.6%) | 0.480 | 21 (25.3%) | 0.855 | 20 (29.9%) | 0.109 |
| 30–39 | 54 (17.9%) | 38 (20.4%) | | 21 (15.8%) | | 13 (15.7%) | | 15 (22.4%) | |
| 40–49 | 59 (19.5%) | 41 (22.0%) | | 23 (17.3%) | | 18 (21.7%) | | 16 (23.9%) | |
| 50–59 | 46 (15.2%) | 29 (15.6%) | | 18 (13.5%) | | 14 (16.9%) | | 7 (10.5%) | |
| 60 and over | 72 (23.8%) | 33 (17.7%) | | 33 (24.8%) | | 17 (20.5%) | | 9 (13.4%) | |
| **State** | | | | | | | | | |
| New South Wales | 88 (29.1%) | 52 (28.0%) | 0.095 | 39 (29.3%) | 0.474 | 22 (26.5%) | 0.113 | 16 (23.9%) | 0.227 |
| Victoria | 76 (25.2%) | 43 (23.1%) | | 34 (25.6%) | | 23 (27.7%) | | 16 (23.9%) | |
| Queensland | 84 (27.8%) | 59 (31.7%) | | 30 (22.6%) | | 27 (32.5%) | | 24 (35.8%) | |
| South Australia | 24 (8.0%) | 16 (8.6%) | | 16 (12.0%) | | 2 (2.4%) | | 2 (3.0%) | |
| Western Australia | 21 (7.0%) | 10 (5.4%) | | 9 (6.8%) | | 8 (9.6%) | | 7 (10.5%) | |
| Tasmania | 4 (1.3%) | 2 (1.1%) | | 1 (0.8%) | | 0 (0.0%) | | 1 (1.5%) | |
| Australian Capital Territory | 5 (1.7%) | 4 (2.2%) | | 4 (3.0%) | | 1 (1.2%) | | 1 (1.5%) | |
| **Managing financially** | | | | | | | | | |
| It is impossible | 9 (3.0%) | 6 (3.2%) | 0.998 | 4 (3.0%) | 0.224 | 3 (3.6%) | 0.829 | 3 (4.5%) | 0.687 |
| It is difficult all of the time | 56 (18.5%) | 33 (17.7%) | | 30 (22.6%) | | 17 (20.5%) | | 15 (22.4%) | |
| It is difficult some of the time | 114 (37.8%) | 70 (37.6%) | | 40 (30.1%) | | 32 (38.6%) | | 24 (35.8%) | |
| It is not too bad | 101 (33.4%) | 64 (34.4%) | | 48 (36.1%) | | 24 (28.9%) | | 19 (28.4%) | |
| It is easy | 22 (7.3%) | 13 (7.0%) | | 11 (8.3%) | | 7 (8.4%) | | 6 (9.0%) | |
| **Education level** | | | | | | | | | |
| Up to year 10 | 42 (13.9%) | 25 (13.4%) | 0.989 | 19 (14.3%) | 0.687 | 10 (12.1%) | 0.847 | 9 (13.4%) | 0.365 |
| Year 12 or equivalent | 51 (16.9%) | 28 (15.1%) | | 24 (18.1%) | | 11 (13.3%) | | 11 (16.4%) | |
| Trade/Vocational | 105 (34.8%) | 67 (36.0%) | | 43 (32.3%) | | 30 (36.1%) | | 18 (26.9%) | |
| University degree | 104 (34.4%) | 66 (35.5%) | | 47 (35.3%) | | 32 (38.6%) | | 29 (43.3%) | |
| **Employment status** | | | | | | | | | |
| Full time work | 101 (33.4%) | 69 (37.1%) | 0.351 | 50 (37.6%) | 0.279 | 25 (30.1%) | 0.220 | 28 (41.8%) | 0.025 (0.167) |
| Part time work | 64 (21.2%) | 43 (23.1%) | | 23 (17.3%) | | 15 (18.1%) | | 16 (23.9%) | |
| Casual/temporary work | 21 (7.0%) | 14 (7.5%) | | 6 (4.5%) | | 10 (12.1%) | | 5 (7.5%) | |
| Looking for work | 21 (7.0%) | 13 (7.0%) | | 9 (6.8%) | | 8 (9.6%) | | 8 (11.9%) | |
| Not in paid workforce | 95 (31.5%) | 47 (25.3%) | | 45 (33.8%) | | 25 (30.1%) | | 10 (14.9%) | |
| **Relationship status** | | | | | | | | | |
| Never married | 79 (26.2%) | 50 (26.9%) | 0.706 | 41 (30.8%) | 0.446 | 21 (25.3%) | 0.315 | 23 (34.3%) | 0.171 |
| Married | 142 (47.0%) | 88 (47.3%) | | 54 (40.6%) | | 43 (51.8%) | | 25 (37.3%) | |
| De facto (opposite sex) | 27 (8.9%) | 17 (9.1%) | | 11 (8.3%) | | 3 (3.6%) | | 9 (13.4%) | |
| De facto (same sex) | 4 (1.3%) | 4 (2.2%) | | 1 (0.8%) | | 1 (1.2%) | | 2 (3.0%) | |
| Separated/divorced/ widowed | 50 (16.6%) | 27 (14.5%) | | 26 (19.6%) | | 15 (18.1%) | | 8 (11.9%) | |
| **PHI status** | | | | | | | | | |

(*Continued*)

**Table 1.** (Continued)

| | Total sample n = 302 (100.0%) | Massage n = 186 (61.6%) | p-value[†] | Chiropractic n = 133 (44.0%) | p-value[†] | Acupuncture n = 83 (27.5%) | p-value[†] | Naturopathy n = 67 (22.2%) | p-value[†] |
|---|---|---|---|---|---|---|---|---|---|
| Has Private health insurance | 186 (61.6%) | 114 (61.3%) | 0.983 | 90 (67.7%) | 0.067 | 55 (66.3%) | 0.301 | 44 (65.7%) | 0.426 |
| Private health insurance covers any CM | 144 (47.7%) | 91 (48.9%) | 0.839 | 72 (54.1%) | 0.104 | 46 (55.4%) | 0.149 | 35 (52.2%) | 0.488 |
| **Health care card status** | 177 (58.6%) | 102 (54.8%) | 0.218 | 82 (61.7%) | 0.310 | 55 (66.3%) | 0.091 | 42 (62.7%) | 0.405 |

*Note.* Some respondents consulted multiple practitioners from more than one profession.

[†]Chi-square test with Cramer's V for significant results, comparing respondents who did and did not consult with this type of complementary medicine practitioner.

*asked me about my use of CM* (mean = 3.48, SD = 1.14) and *They have a good attitude towards CM* (mean = 3.57, SD = 0.87). Compared to those consulting other CM professions, those who had consulted a naturopath had a significantly lower mean for the item *They have my best interests at heart* (*p = 0.005*), while those who had consulted a chiropractor had a significantly higher mean for *They asked me about my use of complementary and alternative medicine* (*p = 0.017*).

Table 4 reports responses by participants who completed CMDI items regarding reasons for non-disclosure of CM use to MDs (n = 87). Means for non-disclosure items were notably lower than those for disclosure. The items attracting the most agreement were *They did not ask me about my CM use* (mean = 3.70, SD = 1.02) and *Complementary and alternative medicines are safe* (mean = 3.26, SD = 0.90). The items attracting the lowest mean scores were *It is none of their business* (mean = 2.77, SD = 0.96) and *I previously had a negative experience when I disclosed using CM* (mean = 2.80, SD = 1.11).

**Table 2. Rates of disclosure behaviour types to complementary medicine practitioners, GPs and specialist doctors, including differences in type of disclosure to GPs and specialist doctors between complementary medicine professions consulted.**

| | Complementary medicine profession consulted | | | |
|---|---|---|---|---|
| | Massage (n = 186) | Chiropractic (n = 133) | Acupuncture (n = 83) | Naturopathy (n = 67) |
| **Conventional medicine use disclosure behaviour to complementary medicine practitioner** | | | | |
| Disclosed ALL | 73 (41.2%) | 67 (56.8%) | 35 (46.7%) | 33 (56.9%) |
| Disclosed SOME | 41 (23.2%) | 25 (21.2%) | 23 (30.7%) | 19 (32.8%) |
| Did NOT disclose | 63 (35.6%) | 26 (22.0%) | 17 (22.7%) | 6 (10.3%) |
| **Complementary medicine use disclosure behaviour to GP** | | | | |
| Disclosed ALL | 128 (70.0%) | 88 (68.8%) | 52 (62.7%) | 42 (62.7%) |
| Disclosed SOME | 36 (19.7%) | 26 (20.3%) | 21 (25.3%) | 19 (28.4%) |
| Did NOT disclose | 19 (10.38%) | 14 (10.9%) | 10 (12.1%) | 6 (9.0%) |
| **p-value[†]** | 0.822 | 0.803 | 0.142 | 0.066 |
| **Complementary medicine use disclosure behaviour to a specialist doctor** | | | | |
| Disclosed ALL | 112 (74.7%) | 89 (79.5%) | 51 (68.0%) | 48 (77.4%) |
| Disclosed SOME | 25 (16.7%) | 13 (11.6%) | 15 (20.0%) | 9 (14.5%) |
| Did NOT disclose | 13 (8.7%) | 10 (8.9%) | 9 (12.0%) | 5 (8.1%) |
| **p-value[†]** | 0.466 | 0.299 | 0.238 | 0.807 |

[†]Chi-square test comparing disclosure behaviour of respondents who did or did not consult with each type of complementary medicine practitioner.

**Table 3. T-test showing differences in reasons for disclosure of complementary medicine use to medical doctors for each type of complementary medicine professional consulted.**

| CMDI disclosure items: | Relative importance of reasons by type of complementary medicine professional consulted (Mean ± SD) | | | | | | | | |
|---|---|---|---|---|---|---|---|---|---|
| Reasons for disclosure of complementary medicine use to a medical doctor | Total sample[†] (n = 263) | Acupuncture (n = 73) | p-value | Chiropractic (n = 114) | p-value | Massage (n = 164) | p-value | Naturopathy (n = 61) | p-value |
| I wanted them to fully understand my health status | 4.44 ± 0.73 | 4.38 ± 0.74 | 0.432 | 4.42 ± 0.81 | 0.699 | 4.45 ± 0.69 | 0.908 | 4.43 ± 0.74 | 0.857 |
| I was concerned about drug interactions with the complementary and alternative medicine I was using | 4.20 ± 0.89 | 4.15 ± 0.94 | 0.568 | 4.19 ± 0.93 | 0.892 | 4.16 ± 0.88 | 0.389 | 4.03 ± 0.87 | 0.092 |
| I have a good relationship with them | 4.07 ± 0.89 | 4.01 ± 0.94 | 0.537 | 4.06 ± 0.90 | 0.911 | 4.10 ± 0.81 | 0.409 | 3.89 ± 0.93 | 0.066 |
| I felt comfortable discussing complementary and alternative medicine with them | 4.07 ± 0.93 | 4.10 ± 0.92 | 0.767 | 4.04 ± 0.96 | 0.612 | 4.10 ± 0.85 | 0.536 | 3.87 ± 1.06 | 0.056 |
| They have my best interests at heart | 4.06 ± 0.82 | 4.03 ± 0.91 | 0.716 | 4.09 ± 0.83 | 0.594 | 4.06 ± 0.76 | 0.923 | 3.80 ± 0.85 | 0.005 |
| I thought they could help with my treatment decisions | 3.94 ± 0.82 | 3.93 ± 0.82 | 0.888 | 4.01 ± 0.84 | 0.253 | 3.91 ± 0.77 | 0.469 | 3.85 ± 0.87 | 0.323 |
| I knew they would be willing to discuss my complementary and alternative medicine use | 3.88 ± 0.92 | 3.86 ± 0.95 | 0.835 | 3.89 ± 0.92 | 0.953 | 3.90 ± 0.85 | 0.759 | 3.72 ± 1.05 | 0.160 |
| They understand my treatment goals | 3.91 ± 0.83 | 3.90 ± 0.77 | 0.919 | 3.94 ± 0.83 | 0.658 | 3.98 ± 0.75 | 0.135 | 3.89 ± 0.86 | 0.770 |
| I thought they might know something about complementary and alternative medicine | 3.84 ± 0.91 | 3.89 ± 0.95 | 0.608 | 3.85 ± 0.94 | 0.916 | 3.82 ± 0.91 | 0.534 | 3.85 ± 0.96 | 0.935 |
| They are open-minded | 3.83 ± 0.88 | 3.84 ± 0.90 | 0.905 | 3.90 ± 0.85 | 0.208 | 3.85 ± 0.87 | 0.596 | 3.66 ± 0.91 | 0.087 |
| I wanted their advice about complementary and alternative medicines | 3.78 ± 0.88 | 3.86 ± 0.85 | 0.341 | 3.86 ± 0.91 | 0.197 | 3.77 ± 0.83 | 0.792 | 3.66 ± 0.89 | 0.211 |
| I knew they would understand about my complementary and alternative medicine use | 3.71 ± 0.98 | 3.78 ± 0.96 | 0.450 | 3.73 ± 0.96 | 0.763 | 3.71 ± 0.94 | 0.895 | 3.52 ± 1.03 | 0.096 |
| I wanted their approval of my complementary and alternative medicine use | 3.62 ± 0.99 | 3.60 ± 0.98 | 0.863 | 3.68 ± 0.98 | 0.425 | 3.62 ± 0.93 | 0.963 | 3.62 ± 1.00 | 0.977 |
| They support my use of complementary and alternative medicines | 3.60 ± 0.86 | 3.70 ± 0.91 | 0.274 | 3.68 ± 0.97 | 0.260 | 3.59 ± 0.81 | 0.761 | 3.70 ± 0.88 | 0.301 |
| They have a good attitude towards complementary and alternative medicine | 3.57 ± 0.87 | 3.63 ± 0.94 | 0.519 | 3.67 ± 0.95 | 0.132 | 3.57 ± 0.82 | 0.981 | 3.44 ± 0.85 | 0.178 |
| They asked me about my use of complementary and alternative medicine | 3.48 ± 1.14 | 3.51 ± 1.07 | 0.780 | 3.67 ± 1.07 | 0.017 | 3.53 ± 1.10 | 0.311 | 3.39 ± 1.24 | 0.522 |

[†] Total sample includes participants who reported full disclosure (Disclosed ALL) or partial disclosure (Disclosed SOME) of complementary medicine use to a medical doctor.

*Note.* Independent t-test analyses compare responses from individuals who did and who did not report consulting with each individual type of complementary medicine profession examined.

Amongst those who had consulted a naturopath, compared to those consulting other CM practitioners, means were significantly higher for items *They do not approve of my use of complementary and alternative medicine* (p = 0.003), *I previously had a negative experience when I disclosed using complementary and alternative medicine* (p = 0.005), *I did not think they would understand my choice* (p = 0.012) and *I did not think they would know anything about complementary and alternative medicine* (p = 0.02). Compared to respondents consulting other CM professions, those who had consulted an acupuncturist produced a significantly higher mean score for item *I felt uncomfortable discussing it with them* (p = 0.018), while for those who had consulted a massage therapist, means were significantly lower for items *I did not think they would understand my choice* (p = 0.003), *It is none of their business* (p = 0.016) and *There was not enough time in the consultation* (p = 0.021).

**Table 4. T-test showing differences in reasons for non-disclosure of complementary medicine use to medical doctors for each type of complementary medicine professional consulted.**

| CMDI non-disclosure items: | Relative importance of reasons by type of complementary medicine professional consulted (Mean ± SD) | | | | | | | | |
|---|---|---|---|---|---|---|---|---|---|
| Reasons for non-disclosure of complementary medicine use to a medical doctor | Total sample[†] (n = 87) | Acupuncture (n = 31) | p-value | Chiropractic (n = 40) | p-value | Massage (n = 55) | p-value | Naturopathy (n = 25) | p-value |
| They did not ask me about my complementary and alternative medicine use | 3.70 ± 1.02 | 3.68 ± 1.08 | 0.873 | 3.73 ± 1.04 | 0.843 | 3.55 ± 1.05 | 0.063 | 3.84 ± 0.94 | 0.425 |
| Complementary and alternative medicines are safe | 3.26 ± 0.90 | 3.35 ± 0.88 | 0.486 | 3.18 ± 0.96 | 0.394 | 3.18 ± 0.86 | 0.262 | 3.36 ± 0.70 | 0.530 |
| I was worried they wouldn't support my treatment decisions | 3.22 ± 0.97 | 3.42 ± 0.76 | 0.117 | 3.23 ± 0.80 | 0.953 | 3.15 ± 1.04 | 0.361 | 3.44 ± 0.82 | 0.177 |
| I did not think they would understand my choice | 3.22 ± 1.00 | 3.26 ± 1.03 | 0.786 | 3.25 ± 0.93 | 0.788 | 2.98 ± 1.01 | 0.003 | 3.64 ± 0.95 | 0.012 |
| I was worried they would judge me | 3.15 ± 1.13 | 3.23 ± 1.06 | 0.641 | 3.20 ± 1.04 | 0.698 | 3.04 ± 1.10 | 0.222 | 3.28 ± 0.98 | 0.461 |
| There was not enough time in the consultation | 3.15 ± 0.99 | 3.03 ± 0.91 | 0.417 | 3.30 ± 0.97 | 0.194 | 2.96 ± 0.96 | 0.021 | 3.24 ± 1.09 | 0.593 |
| I was worried they would discourage my use of complementary and alternative medicine | 3.15 ± 0.99 | 3.13 ± 0.99 | 0.888 | 3.23 ± 0.89 | 0.516 | 3.00 ± 0.98 | 0.066 | 3.40 ± 0.82 | 0.137 |
| I felt uncomfortable discussing it with them | 3.14 ± 1.02 | 3.48 ± 0.93 | 0.018 | 3.30 ± 0.91 | 0.175 | 3.00 ± 1.02 | 0.100 | 3.32 ± 0.95 | 0.295 |
| They did not need to know | 3.13 ± 1.00 | 3.03 ± 0.84 | 0.485 | 3.20 ± 1.04 | 0.529 | 3.02 ± 0.99 | 0.186 | 3.16 ± 0.94 | 0.843 |
| I did not think they would know anything about complementary and alternative medicine | 3.10 ± 1.07 | 3.35 ± 1.11 | 0.103 | 3.03 ± 1.00 | 0.530 | 3.07 ± 1.00 | 0.727 | 3.52 ± 1.12 | 0.020 |
| I was worried they would respond negatively | 3.07 ± 1.00 | 3.13 ± 0.92 | 0.679 | 3.15 ± 1.08 | 0.488 | 2.93 ± 0.98 | 0.082 | 3.36 ± 0.76 | 0.084 |
| They do not approve of my use of complementary and alternative medicine | 3.00 ± 0.98 | 3.16 ± 0.90 | 0.254 | 2.95 ± 0.93 | 0.662 | 2.93 ± 0.94 | 0.366 | 3.48 ± 0.77 | 0.003 |
| I previously had a negative experience when I disclosed using complementary and alternative medicine | 2.80 ± 1.11 | 3.00 ± 0.97 | 0.198 | 2.80 ± 1.09 | 0.972 | 2.67 ± 1.04 | 0.147 | 3.32 ± 0.90 | 0.005 |
| It is none of their business | 2.77 ± 0.96 | 2.84 ± 1.00 | 0.623 | 2.80 ± 0.99 | 0.791 | 2.58 ± 0.88 | 0.016 | 2.72 ± 0.79 | 0.759 |

[†]Total sample includes participants who reported non-disclosure (Did NOT disclose) or partial disclosure (Disclosed SOME) of complementary medicineuse to a medical doctor.

*Note.* Independent t-test analyses compare responses from individuals who did and who did not report consulting with each individual type of complementary medicineprofession examined.

## Reasons for disclosure and non-disclosure of conventional medicine use to CM practitioners

Amongst participants who responded to items regarding disclosure of conventional medicines to CM practitioners (n = 216), the item attracting the highest agreement was *I wanted them to fully understand my health status* (mean = 4.26, SD = 0.79), followed by *They have my best interests at heart* (mean = 3.95, SD = 0.90) and *They understand my treatment goals* (mean = 3.94, SD = 0.82). The item attracting the lowest mean was *I wanted their approval of my conventional medicine use* (mean = 3.22, SD = 1.03). Significantly lower means were seen for item *They are open-minded* amongst respondents who had consulted an acupuncturist (*p = 0.05*) or a naturopath (*p = 0.043*), as well as for item *I wanted them to fully understand my health status* amongst those who had consulted a massage therapist (*p = 0.031*). Significantly higher means were seen for item *I was concerned about drug interactions with the conventional medicine I was using* for those who had consulted a naturopath (*p = 0.039*), and for item *I knew they would understand about my conventional medicine use* amongst those who had consulted a chiropractor (*p = 0.033*). See Table 5.

**Table 5. T-test showing differences in reasons for disclosure of conventional medicine use to complementary medicine practitioners for each type of complementary medicine professional consulted.**

| CONMED-DI disclosure items: | Relative importance of reasons by type of complementary medicine professional consulted (Mean ± SD) | | | | | | | | |
|---|---|---|---|---|---|---|---|---|---|
| Reasons for disclosure of conventional medicines to complementary medicine practitioner | Total sample[†] (n = 216) | Acupuncture (n = 67) | p-value | Chiropractic (n = 104) | p-value | Massage (n = 132) | p-value | Naturopathy (n = 61) | p-value |
| I wanted them to fully understand my health status | 4.26 ± 0.79 | 4.15 ± 0.86 | 0.173 | 4.28 ± 0.78 | 0.728 | 4.17 ± 0.80 | 0.031 | 4.28 ± 0.69 | 0.822 |
| They have my best interests at heart | 3.95 ± 0.90 | 3.84 ± 0.86 | 0.199 | 3.89 ± 0.93 | 0.353 | 3.94 ± 0.85 | 0.771 | 3.90 ± 0.93 | 0.596 |
| They understand my treatment goals | 3.94 ± 0.82 | 3.93 ± 0.77 | 0.820 | 3.94 ± 0.86 | 0.971 | 3.92 ± 0.75 | 0.652 | 3.93 ± 0.65 | 0.899 |
| I was concerned about drug interactions with the conventional medicine I was using | 3.87 ± 0.95 | 3.97 ± 0.89 | 0.300 | 3.83 ± 0.98 | 0.517 | 3.92 ± 0.88 | 0.368 | 4.08 ± 0.86 | 0.039 |
| I felt comfortable discussing conventional medicines with them | 3.85 ± 0.89 | 3.70 ± 0.92 | 0.097 | 3.88 ± 0.95 | 0.714 | 3.80 ± 0.88 | 0.315 | 3.89 ± 0.93 | 0.731 |
| I have a good relationship with them | 3.84 ± 0.95 | 3.91 ± 0.97 | 0.483 | 3.93 ± 0.97 | 0.181 | 3.80 ± 0.95 | 0.445 | 3.87 ± 0.97 | 0.800 |
| They are open-minded | 3.82 ± 0.92 | 3.64 ± 0.93 | 0.050 | 3.85 ± 0.89 | 0.734 | 3.80 ± 0.91 | 0.567 | 3.62 ± 0.90 | 0.043 |
| I knew they would be willing to discuss my conventional medicine use | 3.81 ± 0.86 | 3.78 ± 0.83 | 0.737 | 3.87 ± 0.81 | 0.325 | 3.72 ± 0.86 | 0.066 | 3.93 ± 0.85 | 0.169 |
| They asked me about my use of conventional medicines | 3.75 ± 0.97 | 3.81 ± 0.93 | 0.538 | 3.76 ± 0.97 | 0.835 | 3.73 ± 0.96 | 0.842 | 3.87 ± 0.83 | 0.200 |
| I thought they might know something about conventional medicines | 3.71 ± 0.91 | 3.82 ± 0.76 | 0.182 | 3.68 ± 0.96 | 0.690 | 3.71 ± 0.89 | 0.939 | 3.87 ± 0.76 | 0.074 |
| I thought they could help with my treatment decisions | 3.68 ± 0.92 | 3.61 ± 0.87 | 0.496 | 3.68 ± 0.94 | 0.918 | 3.65 ± 0.88 | 0.627 | 3.70 ± 0.80 | 0.773 |
| They have a good attitude towards conventional medicine | 3.67 ± 0.90 | 3.58 ± 0.91 | 0.355 | 3.76 ± 0.93 | 0.144 | 3.62 ± 0.84 | 0.353 | 3.49 ± 0.85 | 0.073 |
| I knew they would understand about my conventional medicine use | 3.65 ± 0.87 | 3.64 ± 0.81 | 0.943 | 3.78 ± 0.86 | 0.033 | 3.61 ± 0.87 | 0.375 | 3.64 ± 0.84 | 0.926 |
| They support my use of conventional medicines | 3.63 ± 0.88 | 3.61 ± 0.80 | 0.843 | 3.71 ± 0.84 | 0.188 | 3.58 ± 0.87 | 0.260 | 3.54 ± 0.72 | 0.301 |
| I was concerned about side-effects of conventional medicines | 3.55 ± 0.99 | 3.67 ± 0.94 | 0.212 | 3.50 ± 1.01 | 0.508 | 3.61 ± 0.98 | 0.210 | 3.66 ± 1.00 | 0.308 |
| I wanted their advice about conventional medicines | 3.49 ± 0.94 | 3.52 ± 0.88 | 0.706 | 3.53 ± 0.93 | 0.523 | 3.50 ± 0.92 | 0.787 | 3.67 ± 0.93 | 0.069 |
| I wanted their approval of my conventional medicine use | 3.22 ± 1.03 | 3.19 ± 1.02 | 0.823 | 3.30 ± 0.97 | 0.270 | 3.22 ± 1.03 | 0.970 | 3.21 ± 1.03 | 0.968 |

[†]Total sample includes participants who reported full disclosure (Disclosed ALL) or partial disclosure (Disclosed SOME) of conventional medicine use to a complementary medicinepractitioner.

*Note.* Independent t-test analyses compare responses from individuals who did and who did not report consulting with each individual type of complementary medicine profession examined.

For responses regarding non-disclosure of conventional medicines to CM practitioners (n = 172), the highest mean recorded was for *They did not ask me about my conventional medicine use* (mean = 3.40, SD = 0.97), followed by *I did not think it was important* (mean = 3.19, SD = 1.00). Items attracting the lowest mean were *I previously had a negative experience when I disclosed using conventional medicine* (mean = 2.71, SD = 0.96), followed by *I was worried they wouldn't support my treatment decisions* (mean = 2.80, SD = 0.93) and *I was worried they would judge me* (mean = 2.80, SD = 1.01).

Amongst respondents who consulted a naturopath, significantly higher means were recorded for items *I previously had a negative experience when I disclosed using conventional medicine* (p = 0.013) and *They do not approve of my use of conventional medicines* (p = 0.016), while a lower mean was recorded for *I did not think it was important* (p = 0.037). For

**Table 6. T-test showing differences in reasons for non-disclosure of conventional medicine use to medical doctors for each type of complementary medicine professional consulted.**

| CONMED-DI non-disclosure items: | Relative importance of reasons by type of complementary medicine professional consulted (Mean ± SD) | | | | | | | | |
|---|---|---|---|---|---|---|---|---|---|
| Reasons for non-disclosure of pharmaceutical medicines to complementary medicine practitioner | Total sample[†] (n = 171) | Acupuncture (n = 52) | p-value | Chiropractic (n = 66) | p-value | Massage (n = 126) | p-value | Naturopathy (n = 43) | p-value |
| They did not ask me about my conventional medicine use | 3.40 ± 0.97 | 3.29 ± 1.00 | 0.330 | 3.41 ± 0.96 | 0.903 | 3.40 ± 0.95 | 0.985 | 3.23 ± 1.07 | 0.197 |
| I did not think it was important | 3.19 ± 1.00 | 3.15 ± 1.04 | 0.813 | 3.20 ± 1.01 | 0.871 | 3.21 ± 1.01 | 0.584 | 2.91 ± 0.97 | 0.037 |
| They did not need to know | 3.10 ± 1.01 | 2.90 ± 0.96 | 0.094 | 3.14 ± 1.02 | 0.706 | 3.08 ± 1.02 | 0.665 | 2.88 ± 1.16 | 0.106 |
| I did not think they would understand my choice | 2.97 ± 0.97 | 2.98 ± 0.94 | 0.929 | 3.08 ± 1.07 | 0.283 | 2.94 ± 0.96 | 0.443 | 3.14 ± 1.13 | 0.239 |
| There was not enough time in the consultation | 2.95 ± 0.98 | 2.79 ± 0.87 | 0.147 | 3.00 ± 1.04 | 0.623 | 2.87 ± 0.93 | 0.049 | 2.98 ± 0.96 | 0.856 |
| I was worried they would discourage my use of conventional medicine | 2.93 ± 0.95 | 2.94 ± 0.94 | 0.910 | 3.03 ± 0.94 | 0.274 | 2.83 ± 0.93 | 0.016 | 3.07 ± 0.86 | 0.265 |
| I forgot to mention it | 2.92 ± 0.98 | 2.69 ± 1.00 | 0.041 | 2.94 ± 1.02 | 0.871 | 2.95 ± 0.95 | 0.529 | 2.67 ± 0.81 | 0.054 |
| I do not use conventional medicines regularly enough | 2.90 ± 0.98 | 2.92 ± 0.97 | 0.884 | 3.02 ± 1.06 | 0.253 | 2.82 ± 0.94 | 0.048 | 2.84 ± 0.84 | 0.595 |
| I did not think they would know anything about conventional medicine | 2.88 ± 0.97 | 2.75 ± 1.03 | 0.237 | 2.91 ± 0.99 | 0.781 | 2.87 ± 0.98 | 0.822 | 2.86 ± 0.91 | 0.861 |
| I felt uncomfortable discussing it with them | 2.88 ± 0.97 | 2.90 ± 1.00 | 0.853 | 3.06 ± 0.94 | 0.057 | 2.83 ± 0.94 | 0.194 | 3.07 ± 1.12 | 0.145 |
| I was worried they would respond negatively | 2.85 ± 0.92 | 2.75 ± 0.79 | 0.359 | 2.95 ± 0.92 | 0.231 | 2.79 ± 0.94 | 0.198 | 3.00 ± 0.87 | 0.212 |
| They do not approve of my use of conventional medicines | 2.84 ± 0.95 | 2.88 ± 1.02 | 0.663 | 3.02 ± 1.03 | 0.052 | 2.73 ± 0.91 | 0.015 | 3.14 ± 1.10 | 0.016 |
| It is none of their business | 2.82 ± 0.99 | 2.81 ± 0.99 | 0.995 | 2.80 ± 1.03 | 0.967 | 2.71 ± 0.94 | 0.041 | 2.70 ± 1.08 | 0.407 |
| I was worried they wouldn't support my treatment decisions | 2.80 ± 0.93 | 2.92 ± 0.97 | 0.259 | 2.97 ± 0.93 | 0.060 | 2.71 ± 0.89 | 0.025 | 3.00 ± 1.00 | 0.106 |
| I was worried they would judge me | 2.80 ± 1.01 | 2.79 ± 0.98 | 0.914 | 2.92 ± 1.01 | 0.207 | 2.78 ± 0.96 | 0.614 | 3.02 ± 1.12 | 0.096 |
| I previously had a negative experience when I disclosed using conventional medicine | 2.71 ± 0.96 | 2.63 ± 1.01 | 0.514 | 2.82 ± 1.02 | 0.235 | 2.61 ± 0.89 | 0.028 | 3.02 ± 0.99 | 0.013 |

[†]Total sample includes participants who reported non-disclosure (Did NOT disclose) or partial disclosure (Disclosed SOME) of conventional medicine use to a complementary medicine practitioner.

*Note*. Independent t-test analyses compare responses from individuals who did and who did not report consulting with each individual type of complementary medicine profession examined.

respondents who had consulted an acupuncturist, a significantly lower mean was recorded for *I forgot to mention it* (*p = 0.041*). Lower means were seen amongst respondents who had consulted a massage therapist for a number of items, namely *They do not approve of my conventional medicine use* (*p = 0.015*), *I was worried they would discourage my use of conventional medicine* (*p = 0.016*), *I was worried they wouldn't support my treatment decisions* (*p = 0.025*), *I previously had a negative experience when I disclosed using conventional medicine* (*p = 0.028*), *It is none of their business* (*p = 0.041*), *I do not use conventional medicines regularly enough* (*p = 0.048*), and *There was not enough time in the consultation* (*p = 0.049*). See Table 6.

## Discussion

This study is the first to examine disclosure of both CM and conventional medicine use to health professionals by patients with chronic conditions across a range of conventional medicine and CM contexts. Our findings indicate that rates of disclosure of CM use to MDs by those with chronic conditions appear much higher than previous estimates of disclosure in the general population [11], while rates of disclosure of conventional medicine use to CM

practitioners may be concerningly low. The patients with chronic conditions in our study choose to disclose primarily due to a desire to have their health status understood by their care providers, and fail to disclose primarily due to a lack of inquiry from care providers.

The finding that disclosure rates to MDs appear higher than disclosure rates to CM practitioners is noteworthy, considering some of the most highly ranked reasons for disclosing to CM practitioners suggest a respectful, communicative patient-practitioner relationship (e.g. *They have my best interests at heart* and *They understand my treatment goals*). Patient-practitioner communication in CM clinical settings is facilitated by longer consultation times, empathic, person-centered approaches by CM practitioners, and the holistic philosophies underlying CM practice [17]. In contrast, patient-provider communication in conventional medical settings is reportedly limited by shorter consultation times, barriers to continuity of care, and a less person-centered experience for patients [18, 19]. Yet, our results suggest disclosure may be facilitated by factors beyond consultation time or general person-centered, holistic approaches to care and communication. Robust comparisons of disclosure rates between complementary and conventional medicine settings have thus far been inhibited by a paucity of research examining disclosure of conventional medicine use to CM practitioners. Our findings are closely aligned with those of a study which briefly compared rates of disclosure between those consulting MDs and those consulting CM practitioners [13], while another study found substantially higher disclosure rates amongst those disclosing to naturopaths compared to MDs [12].

Indeed, disclosure to naturopaths was highest amongst the CM professions consulted in our study. This finding may reflect the differences in practice and treatment across different CM professions; while the massage, chiropractic and acupuncture professions most commonly use non-ingested treatments (manual therapy or acupuncture needles), naturopathic practitioners frequently prescribe orally-ingested herbs, supplements and therapeutic foods which can present a greater risk of interaction with conventional/pharmaceutical medicines [20, 21]. Patients accessing naturopathic care may be aware of this risk given those participants who consulted a naturopath in our study were more likely to report a stronger degree of concern about drug interactions as a reason for disclosing to the naturopath. Our study also found non-disclosure of conventional medicine use to naturopaths was associated with reports of negative previous experiences of disclosing and a patient belief that naturopaths do not approve of conventional medicine use. Previous studies have highlighted similar experiences and perspectives amongst patients regarding disclosure of CM use to MDs and other conventional medical providers [11]. While no previous literature has examined such patient experiences or perspectives regarding disclosure to naturopaths, research has identified a diversity and complexity of views amongst CM practitioners toward conventional medicines, such as vaccines [22], and that naturopaths typically hold supportive views regarding the integration of conventional medicines and complementary medicine generally [23]. Disclosure rates in naturopathic practice might be improved by ensuring supportive communication by naturopaths to patients' concomitant use of naturopathy and conventional medicine.

The finding that the lowest rates of disclosure to CM practitioners were amongst those consulting massage therapists in our sample may reflect the way patients use massage therapy and the nature of massage therapy practice. Compared to the other CM professions included in this study, massage has been suggested as more likely to be used as a non-essential/luxury practice rather than being used solely for the treatment or management of a health condition [24]. When used as treatment, massage therapy is primarily accessed for musculoskeletal complaints, rather than for conditions involving additional complex physiological considerations [25] and typically involves a biomechanical focus in the scope of practice [26]. Due to the aspects of perceived luxury and more targeted treatment purposes, disclosure may be seen as

less necessary by patients of massage therapists, particularly as the profession does not typically involve prescription of ingested treatments that may present a risk of drug interaction. However, patient disclosure of conventional medicine use should still be encouraged by massage therapists through patient education in order to ensure a full understanding of the patient's health status and potential contraindications, such as cardiovascular conditions and associated pharmaceutical treatments that may carry risk of bruising, bleeding or blood clots [27].

While the primary reason for non-disclosure to MDs reported by our participants was not being asked by the doctor about CM, those who consulted a chiropractor reported a significantly higher mean for having disclosed due to being questioned about CM by their MD. Additionally, patients of chiropractors also reported a higher mean regarding disclosing to their chiropractor for the item *I knew they would understand about my conventional medicine use.* This may be reflective of the status of chiropractic practice in Australia being treated as an allied health profession, which generates referrals for patients from MDs to chiropractic care and subsequently better integrated communication about concomitant use of conventional and chiropractic care [28]. This may be contrasted with reasons for non-disclosure to MDs given by participants consulting with less integrated CM professions in our study—higher means were reported by naturopathy patients regarding a perception of their doctor not approving of their CM use, as well as having had a negative experience disclosing CM use previously.

## Implications for policy and practice

Our study showed a failure to be asked about CM or conventional medicine use by the consulting care provider was the most prominent reason for non-disclosure to both MDs and CM practitioners, regardless of the CM profession being consulted. This finding is consistent with previous literature on CM use disclosure to MDs and other conventional medical providers [11] and identifies an opportunity for all care providers to improve patient management for those with chronic conditions through simple inquiry. Prior research has demonstrated that disclosure of CM to MDs can be improved through inclusion of a question about CM use in addition to usual clinical case-taking procedures [29], and this may be applicable to CM settings also. In view of participants reporting a desire to have their health status fully understood as a primary reason for disclosing, ensuring that patients are educated about the importance of disclosing other medication and treatment use as part of direct inquiry may also enhance patient-practitioner communication around disclosure. Aligning clinical practice with contemporary health policy relating to chronic condition management, such as recommendations for person-centered and integrated care [1, 30], may foster patient-practitioner relationships and clinical environments which encourage communication around concomitant use of multiple forms of health care.

## Limitations

While our study findings provide a new depth of understanding to an issue integral to the care of patients with chronic conditions, the study is not without limitations. The initial sample was broadly representative of the national general population, however, the online setting and self-report format of the survey may have led to responder and recall bias, limiting generalisability. The use of online research company databases may contribute to such biases as database membership is limited to internet users and may appeal to people with unknown characteristics [31]. In response to this, steps were taken to reduce these biases through purposive elements in sampling, and by limiting questions about consultation and disclosure to the preceding 12 months. Health status regarding chronic conditions was assessed by asking participants if they

had been diagnosed with or treated for a chronic condition within the previous three years. However, the duration of conditions was not ascertained and thus the experiences of participants may not accurately reflect the impact of chronicity. In addition, patient health-related characteristics such as age and number of chronic conditions appear to contribute to disclosure, raising questions for future research surrounding whether complexity or severity of chronic conditions may also play a part.

As some participants had used more than one form of CM, it cannot be determined whether the CM use they had disclosed/not disclosed to their MD was the same as the CM profession they identified as having consulted and disclosed/not disclosed to. It is also likely that reasons for disclosure and non-disclosure are related, rather than independent, and future analyses could involve structural equations to explore such dynamics. Finally, the disclosure indices limited responses regarding reasons to predetermined lists without opportunity for participants to provide open-text responses. Nevertheless, the indices were developed through rigorous examination of existing, expansive literature and the measures were subject to validation analyses [15, 16].

## Conclusion

Communication between individuals with chronic conditions and their health care providers regarding disclosure of complementary and conventional medicine use is influenced by a number of contextual factors relating to the clinical encounter, patient-provider relationship, and patient beliefs. While it is important to patients that their providers have a full understanding of their health status, opportunities to develop such understanding may not be maximised if information regarding various treatments being used by patients fails to be communicated. Disclosure may be better facilitated by patient education regarding the importance of sharing this information with care providers, direct inquiry and supportive approaches to discussion by care providers.

## Supporting information

**S1 File. CAMUHLD survey copy.** Complete questionnaire for the CAMUHLD survey 2017. (PDF)

## Acknowledgments

The authors wish to thank the participants for the contribution of their time and attention.

## Author Contributions

**Conceptualization:** Hope Foley, Amie Steel, Erica McIntyre, Joanna Harnett, David Sibbritt, Jon Adams.

**Data curation:** Hope Foley, Erica McIntyre.

**Formal analysis:** Hope Foley, Amie Steel.

**Funding acquisition:** Amie Steel, Jon Adams.

**Investigation:** Hope Foley, Amie Steel, Erica McIntyre.

**Methodology:** Hope Foley, Amie Steel, Erica McIntyre, Joanna Harnett, David Sibbritt.

**Project administration:** Hope Foley, Amie Steel, Erica McIntyre, Jon Adams.

**Resources:** Erica McIntyre, Jon Adams.

**Supervision:** Amie Steel, Jon Adams.

**Visualization:** Hope Foley, Amie Steel.

**Writing – original draft:** Hope Foley.

**Writing – review & editing:** Hope Foley, Amie Steel, Erica McIntyre, Joanna Harnett, David Sibbritt, Jon Adams.

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
