## [Decision Letter · Decision Letter 0]

3 Sep 2021

PONE-D-21-16947

Disclosure of conventional and complementary medicine use to medical doctors and complementary medicine practitioners: A survey of rates and reasons amongst those with chronic conditions

PLOS ONE

Dear Dr. Foley,

Thank you for submitting your manuscript to PLOS ONE. After careful consideration, we feel that it has merit but does not fully meet PLOS ONE’s publication criteria as it currently stands. Therefore, we invite you to submit a revised version of the manuscript that addresses the points raised during the review process.

We look forward to receiving your revised manuscript.

Kind regards,

Vijayaprakash Suppiah, PhD

Academic Editor

PLOS ONE

Journal Requirements:

2. Please include additional information regarding the survey or questionnaire used in the study and ensure that you have provided sufficient details that others could replicate the analyses. For instance, if you developed a questionnaire as part of this study and it is not under a copyright more restrictive than CC-BY, please include a copy, in both the original language and English, as Supporting Information. If the original language is written in non-Latin characters, for example Amharic, Chinese, or Korean, please use a file format that ensures these characters are visible.

3. Please state whether you validated the questionnaire prior to testing on study participants. Please provide details regarding the validation group within the methods section.

Reviewers' comments:

Reviewer's Responses to Questions

**Comments to the Author**

1. Is the manuscript technically sound, and do the data support the conclusions?

Reviewer #1: Yes

Reviewer #2: Yes

2. Has the statistical analysis been performed appropriately and rigorously? 

Reviewer #1: Yes

Reviewer #2: I Don't Know

3. Have the authors made all data underlying the findings in their manuscript fully available?

Reviewer #1: Yes

Reviewer #2: No

4. Is the manuscript presented in an intelligible fashion and written in standard English?

Reviewer #1: Yes

Reviewer #2: Yes

5. Review Comments to the Author

Reviewer #1: In this study, a questionnaire survey was used to understand the rate and reasons of disclosure and non-disclosure of conventional and complementary medicine use to medical doctors and complementary medicine practitioners amongst patients with chronic conditions. Full disclosure rates were highest for disclosure of CM to specialist doctors. Age and number of diseases were the predictors of full disclosure.

“I wanted them to fully understand my health status” and “They did not ask me about my CM use” were main reasons for disclosure and non-disclosure of CM use to MDs and conventional medicine use to CM practitioners.

1. I noticed that 32.5 percent of all respondents were 18-29 years old. Is it true that prevalence of chronic conditions is higher in younger people than in older people, or is there a response bias?

2. The results showed older patients and patients with a high number of chronic conditions are more likely to disclose all CM use to GPs. Disclosure may be not only associated with the number of chronic conditions, but also with the complexity and severity of the. Can relevant studies be complemented in this study?

3. The reasons for disclosure and non-disclosure of CM use to MDs are not independent of each other. Try using structural equations to explore the reasons of disclosure and non-disclosure.

Reviewer #2: This paper looks fine to me. A few suggestions for minor amendments:

1. Check for typos

2. Add a reference for Qualtrics and for CAMUHLD (and/or add more details to this paper)

3. I am unclear who the database of participants are and why they have volunteered to take part in studies. I wonder if there is a chance they are a particular group of society? I know you try to check they are representative but it would be good to know a bit more about them and to highlight any particular limtations with this group.

4. How was the survey sent out and completed? I presume online, but Methods don't actually say.

5. In the results you use 'predominantly' for a figure of 29% - I would say this was not a majority?

6. Results (and tables) are very long - any way you can make them a bit more concise?

7. In Tables make sure all acronyms are in full (in footnotes maybe) as I believe tables should be able to be understood on their own. In table 1 I'm not sure what VET means.

6. PLOS authors have the option to publish the peer review history of their article (what does this mean?). If published, this will include your full peer review and any attached files.

Reviewer #1: No

Reviewer #2: No

---

## [Author Response · Author response to Decision Letter 0]

10 Sep 2021

Letter of response to Academic Editor and Reviewers:

Response to Academic Editor

Dear Dr Vijayaprakash Suppiah and Editorial Team,

Thank you for your time and attention to our manuscript. In regards to the additional requirements laid out in your email for our revised submission, please see our responses below, followed by our responses to Reviewer comments.

Comments

Response: The manuscript has been checked against the PLOS ONE style requirements and templates and adjusted accordingly, including file names.

2. Please include additional information regarding the survey or questionnaire used in the study and ensure that you have provided sufficient details that others could replicate the analyses. For instance, if you developed a questionnaire as part of this study and it is not under a copyright more restrictive than CC-BY, please include a copy, in both the original language and English, as Supporting Information. If the original language is written in non-Latin characters, for example Amharic, Chinese, or Korean, please use a file format that ensures these characters are visible.

Response: A copy of the survey has now been included as a Supporting Information file.

3. Please state whether you validated the questionnaire prior to testing on study participants. Please provide details regarding the validation group within the methods section.

Response: The manuscript has now been amended to clearly state that validation was undertaken after data collection (page 6). Published results of the validation analyses have been referenced.

4. We note that you have indicated that data from this study are available upon request. PLOS only allows data to be available upon request if there are legal or ethical restrictions on sharing data publicly.

Response: The data is restricted due to specifications laid out in the project’s ethics approval from the ethics committee at the Researchers’ institution. This has been detailed in the revised cover letter.

Response: The reference list has been checked for formatting and typographical errors have been amended. None of the cited articles have been retracted. 

Response to Reviewers

The authors thank the reviewers for the generous contribution of their time and for their considered feedback, which has assisted in strengthening the integrity and value of this manuscript.

Reviewer #1 Comments

1. I noticed that 32.5 percent of all respondents were 18-29 years old. Is it true that prevalence of chronic conditions is higher in younger people than in older people, or is there a response bias?

Response: The Reviewer raises an insightful question. The full sample (n=2,019) taken for the CAMUHLD survey was nationally representative of the Australian population in regards to age (as noted on manuscript page 5, Participants and recruitment section), and chronic condition prevalence was aligned with previous estimates for the Australian population [1]. However, the sub-sample used for analyses in the current paper focussed on those CAMUHLD participants who “had consulted with CM practitioners from one of the professions most commonly accessed by respondents with chronic conditions (massage therapy, chiropractic, acupuncture and naturopathy)” (see manuscript page 6). The age groups of 18-29 and 60-and-over reported higher rates of CM use [1], which is reflected in the higher representation of these age groups in our current analyses. Given the representativeness of the CAMUHLD sample, rates of CM use are a more likely cause than response bias for the higher percentage of younger respondents. Regardless, the possibility of response bias is noted in the Limitations section as follows (page 29):

“The initial sample was broadly representative of the national general population, however, the online setting and self-report format of the survey may have led to responder and recall bias, limiting generalisability. Steps were taken to reduce these biases through purposive elements in sampling, and by limiting questions about consultation and disclosure to the preceding 12 months.”

2. The results showed older patients and patients with a high number of chronic conditions are more likely to disclose all CM use to GPs. Disclosure may be not only associated with the number of chronic conditions, but also with the complexity and severity of the. Can relevant studies be complemented in this study?

Response: Thank you to the Reviewer for raising this interesting point. Despite having reviewed the literature on disclosure prior to conducting the CAMUHLD survey [2], it appears that studies to date have not thoroughly explored associations with complexity or severity of health conditions. While not within the scope of this paper or the CAMUHLD dataset (which did not record complexity or severity), this is a question worthy of future exploration, and the Discussion section has been updated to reflect this as follows (page 29): 

“In addition, patient health-related characteristics such as age and number of chronic conditions appear to contribute to disclosure, raising questions for future research surrounding whether complexity or severity of chronic conditions may also play a part.”

3. The reasons for disclosure and non-disclosure of CM use to MDs are not independent of each other. Try using structural equations to explore the reasons of disclosure and non-disclosure.

Response: We agree with the Reviewer that reasons for disclosure and non-disclosure are not independent of each other. Examination of the relationships between these factors is not within the scope of this paper and is worthy of its own focussed attention in future analyses. The Limitations section has been amended to reflect this as follows (page 29):

“It is also likely that reasons for disclosure and non-disclosure are related, rather than independent, and future analyses could involve structural equations to explore such dynamics.”

 Reviewer #2 Comments

1. Check for typos

Response: The manuscript has been checked and typographical errors corrected.

2. Add a reference for Qualtrics and for CAMUHLD (and/or add more details to this paper)

Response: Thank you to the Reviewer for noting this need for further clarity. Qualtrics is a research recruitment company, which has been made more clear in the manuscript by adding the word “company” to the description in the methods as follows (page 5, changes underlined):

“Survey participants were adult members (aged 18 and over) of the Qualtrics research recruitment company’s database, via which they were invited to participate.”

CAMUHLD (Complementary and Alternative Medicine Use, Health Literacy and Disclosure) is the title of the survey used in the study. The introductory paragraph of the Materials and Methods section has been updated to better describe this, and a reference to the initial publication arising from the survey has been inserted as follows (page 5, changes underlined):

“This paper reports on data collected via the Complementary and Alternative Medicine Use, Health Literacy and Disclosure (CAMUHLD) cross-sectional survey conducted online between 26 July and 28 August 2017 as part of the Complementary and Alternative Medicine Use, Health Literacy and Disclosure (CAMUHLD) project. The survey was administered nationally across Australia [14]. Analyses presented here are nested within the CAMUHLD project utilise data from a sub-set of the CAMUHLD sample.”

3. I am unclear who the database of participants are and why they have volunteered to take part in studies. I wonder if there is a chance they are a particular group of society? I know you try to check they are representative but it would be good to know a bit more about them and to highlight any particular limitations with this group.

Response: Thank you to the Reviewer for these questions. As clarified in the previous response, Qualtrics are an online research recruitment company. The use of such companies for academic research is increasingly common [3] and as with any method of data collection, certain limitations are attached. The Limitations section of the manuscript has been amended to more explicitly address this as follows (page 29, changes underlined):

“The initial sample was broadly representative of the national general population, however, the online setting and self-report format of the survey may have led to responder and recall bias, limiting generalisability. The use of online research company databases may contribute to such biases as database membership is limited to internet users and may appeal to people with unknown characteristics [31]. In response to this, steps were taken to reduce these biases through purposive elements in sampling, and by limiting questions about consultation and disclosure to the preceding 12 months.

4. How was the survey sent out and completed? I presume online, but Methods don't actually say.

Response: Yes, the survey was administered online, as stated in the opening paragraph of the Materials and Methods section (Study Design and Setting section, page 5 – “…cross-sectional survey conducted online between 26 July and 28 August 2017”). This has now been reiterated under the Instrument section to ensure clarity for the reader as follows (page 6, changes underlined):

“The fifty-item CAMUHLD survey, administered online, included domains of socio-demographics, health status, health service utilisation, and health communication.”

5. In the results you use 'predominantly' for a figure of 29% - I would say this was not a majority?

Response: The word ‘predominantly’ has been changed to ‘most commonly’, to reflect that this was the most commonly reported category for the relevant variable (page 9, Participant Characteristics).

6. Results (and tables) are very long - any way you can make them a bit more concise?

Response: The Results have been edited to simplify wording and reduce length. The Tables have been edited to improve clarity (see next comment), however, we feel that it is not possible to reduce them without losing important information.

7. In Tables make sure all acronyms are in full (in footnotes maybe) as I believe tables should be able to be understood on their own. In table 1 I'm not sure what VET means.

Response: Thank you to the Reviewer for noting this. The tables have been updated accordingly in the manuscript document (pages 10, 13, 16, 19, 22, 24).

References:

1. Steel A, McIntyre E, Harnett J, Foley H, Adams J, Sibbritt D, et al. Complementary medicine use in the Australian population: results of a nationally-representative cross-sectional survey. Sci Rep. 2018;8(1):17325. doi: 10.1038/s41598-018-35508-y.

2. Foley H, Steel A, Cramer H, Wardle J, Adams J. Disclosure of complementary medicine use to medical providers: a systematic review and meta-analysis. Sci Rep. 2019;9(1):1-17. doi: 10.1038/s41598-018-38279-8.

3. Schoenherr T, Ellram LM, Tate WL. A Note on the Use of Survey Research Firms to Enable Empirical Data Collection. J Bus Logist. 2015;36(3):288-300. doi: https://doi.org/10.1111/jbl.12092.

---

## [Decision Letter · Decision Letter 1]

8 Oct 2021

Disclosure of conventional and complementary medicine use to medical doctors and complementary medicine practitioners: A survey of rates and reasons amongst those with chronic conditions

PONE-D-21-16947R1

Dear Dr. Foley,

We’re pleased to inform you that your manuscript has been judged scientifically suitable for publication and will be formally accepted for publication once it meets all outstanding technical requirements.

Kind regards,

Vijayaprakash Suppiah, PhD

Academic Editor

PLOS ONE

Reviewer's Responses to Questions

**Comments to the Author**

1. If the authors have adequately addressed your comments raised in a previous round of review and you feel that this manuscript is now acceptable for publication, you may indicate that here to bypass the “Comments to the Author” section, enter your conflict of interest statement in the “Confidential to Editor” section, and submit your "Accept" recommendation.

Reviewer #2: All comments have been addressed

2. Is the manuscript technically sound, and do the data support the conclusions?

Reviewer #2: (No Response)

3. Has the statistical analysis been performed appropriately and rigorously? 

Reviewer #2: (No Response)

4. Have the authors made all data underlying the findings in their manuscript fully available?

Reviewer #2: (No Response)

5. Is the manuscript presented in an intelligible fashion and written in standard English?

Reviewer #2: (No Response)

6. Review Comments to the Author

Reviewer #2: (No Response)

7. PLOS authors have the option to publish the peer review history of their article (what does this mean?). If published, this will include your full peer review and any attached files.

Reviewer #2: **Yes: **Ava Lorenc

---

## [Editor Report · Acceptance letter]

26 Oct 2021

PONE-D-21-16947R1 

Disclosure of conventional and complementary medicine use to medical doctors and complementary medicine practitioners: A survey of rates and reasons amongst those with chronic conditions 

Dear Dr. Foley:

I'm pleased to inform you that your manuscript has been deemed suitable for publication in PLOS ONE. Congratulations! Your manuscript is now with our production department. 

Kind regards, 

on behalf of

Dr. Vijayaprakash Suppiah 

Academic Editor

PLOS ONE